Intraspecific and spatial variation in habitat use by sperm whales (Physeter macrocephalus) along the west coast of Martinique

Laurent Séréna 1
Poupard Marion 1
Ortolé Célia 1
Valin Céline 1
de Montgolfier Benjamin 1 2 3 b.montgolfier@aquasearch.fr
1 Aquasearch , Sainte-Luce , Martinique
2 Bio-Laurentia , Saint-Anaclet, Québec , Canada
3 Institut des Sciences de la Mer, University of Québec at Rimouski , Rimouski, Québec , Canada
Coscarella Mariano
Electronic publication date: 2025 Jul 14
Publication date: 2025
Volume: 13
Electronic Location ID: e19614
Received 2024 Dec 10; Accepted 2025 May 27
Copyright: © 2025 Laurent et al.
Copyright year: 2025
Copyright holder: Laurent et al.
License: This is an open access article distributed under the terms of the Creative Commons Attribution License, which permits unrestricted use, distribution, reproduction and adaptation in any medium and for any purpose provided that it is properly attributed. For attribution, the original author(s), title, publication source (PeerJ) and either DOI or URL of the article must be cited.
License URL: https://creativecommons.org/licenses/by/4.0/

Keywords: Cetacean, Sperm whale, Habitat, Caribbean, Conservation

Funding: Grand Port Maritime de Martinique, the Office Français de la Biodiversité and the Agoa Sanctuary This work was supported by the Grand Port Maritime de Martinique, the Office Français de la Biodiversité and the Agoa Sanctuary. The funders had no role in study design, data collection and analysis, decision to publish, or preparation of the manuscript.

==============================
For a deep-diving cetacean species like the sperm whale, acoustics is a vital tool for research. This need is especially pressing in the eastern Caribbean, where the habitat of marine mammals overlaps with heavy maritime traffic, leading to noise pollution and an increased risk of vessel collisions. To mitigate this risk, understanding their habitat use is essential. Mature males are generally solitary and migrate over long distances, while females and immatures form stable social units in subtropical and tropical waters. In this study, we examined intraspecific variation in distribution and habitat use among individuals along the Caribbean coast of Martinique, using both visual and acoustic data. Over the course of 24 surveys, 19 aggregations involving a total of 74 individual sightings were characterised, recognizing that some individuals may have been recorded multiple times. Using the inter-pulse interval (IPI) of clicks, we estimated individual size, which provided insights into the age and/or sex of each individual. Habitat characteristics included bathymetry, distance from the coast, and seabed slope. Our results on social structure are in line with previous literature: 37% of the aggregations were made up of females and/or juveniles, immatures, with a mature male nearby, with temporal changes in aggregations linked to male migration patterns. Spatial distribution and habitat use appeared consistent across aggregation types, regardless of group size, average individual size, or the presence of immatures. However, specific areas were identified for hunting and socialising based on bathymetry. This study highlights the importance of bathymetry and/or distance from the coast and temporal dynamics related to variations in weather conditions and movements of breeding males, in understanding habitat use by sperm whales in the eastern Caribbean. The lack of observed influence of seabed slope suggests that our spatial scale may have been too limited, or that finer details regarding seabed characteristics are needed. These findings could inform traffic management strategies to reduce the risk of vessel collisions with sperm whales.

Introduction

While standard visual observation methods provide limited perspective on the behaviour of marine animals, passive acoustics offer a way of understanding underwater behaviour, particularly for deep-diving species such as sperm whales (Physeter macrocephalus) (Whitehead, 2003). Acoustics allows underwater species to be studied over larger temporal and spatial scales (Mellinger et al., 2007) and also to account for environmental noise disturbances (Browning et al., 2017). Large cetaceans such as sperm whales are at high risk of collision (Laist et al., 2001; Di-Meglio, David & Monestiez, 2018; Fais et al., 2016) due to their large size and slow speed. Moreover immature individuals dive shallower and for shorter periods than adults, resulting in longer surface intervals and increased exposure to maritime traffic (Tønnesen et al., 2018; Miller, Dawson & Vennell, 2013).

In the Canary Islands, Fais et al. (2016) demonstrated that collisions are a significant cause of mortality in sperm whales. Although Caribbean sperm whales are listed as Vulnerable (Savouré-Soubelet et al., 2016b) and maritime traffic along the west coast of Martinique is high (see Fig. 1), no study has characterised the population of these sperm whales. However, the risk of collision with ships is real. Indeed, an adult humpback whale stranded at Le Lamentin in Martinique, following injuries caused by boats (Escarguel, 2022).

Figure 1 Sperm whale observations since 2013 and acoustic points made along the 1,500 m isobath from south to north during the study.

The density of maritime traffic was given in routes/0.08 km2 inspired by MarineTraffic data for 2024.

Unlike mature males, females, juveniles, and immatures form resident groups that remain in subtropical and tropical waters.(Dufault, Whitehead & Dillon, 1999; Whitehead, 2003). These social units are therefore constantly exposed to anthropogenic activities. Understanding how cetaceans use their habitat could potentially inform marine spatial planning efforts, for example by identifying areas at high risk of collision, as has been done in the northwestern Mediterranean to protect fin whales (Balaenoptera physalus) and sperm whales (Grossi et al., 2021). In response to the risk of collision in the Mediterranean, a collaborative computer system called Repcet® (Real-Time Cetacean Tracking) has been developed to give ships access to the position of cetaceans observed on their route (Arcangeli et al., 2014).

Sperm whales are one of the twenty-four species of cetacean found in the French West Indies (Savouré-Soubelet et al., 2016a). They belong to the suborder odontocetes and exhibit remarkable sexual dimorphism (Rice, 1989). Females generally measure between 9 and 12 m in length and weigh up to 15 tonnes, while males measure up to 20 m in length and weigh between 45 and 57 tonnes (Rice, 1989). Sperm whales can dive to depths of up to 3,000 m and remain submerged for an average of 45 min (Savouré-Soubelet et al., 2016a). In addition to their sexual dimorphism, sperm whales exhibit two behavioural patterns (Dufault, Whitehead & Dillon, 1999; Whitehead, 2003). Social units consisting of females, juveniles and immatures are mostly found in subtropical and tropical regions, while males migrate from high latitudes in cold waters to subtropical and tropical areas to breed (Dufault, Whitehead & Dillon, 1999; Whitehead, 2003). It is also know that adult females in a social unit engage in babysitting—taking turns caring for the young, which affects their hunting behaviour (Arnbom & Whitehead, 1989; Whitehead, 1996; Gero, 2005; Gero et al., 2009). These differences between mature males and social units suggest intraspecific variation in habitat distribution and use.

As odontocetes, sperm whales spend the majority of their time echolocating to hunt and locate their habitat (Whitehead & Weilgart, 1991; Watwood et al., 2006). They produce several types of clicks, defined by click rate, which correspond to different behaviours. “Regular clicks” are series of clicks with inter-click intervals (ICI) of about 0.5 to 2 s, associated with hunting (Whitehead & Weilgart, 1990; Watwood et al., 2006). The “buzzes” or “creaks” are produced during the capture of prey attempts and are characterised by the emission of closely spaced clicks whose interval varies from 0.02 to 0.2 s (Goold & Jones, 1995). Finally, “codas” are stereotypical series of three to 20 clicks, lasting 0.2 to 2 s, emitted during socialisation events (Watkins & Schevill, 1977). Distinct coda dialects characterise vocal clans—assemblages of units that share a similar dialect and may include thousands of individuals. (Rendell & Whitehead, 2003; Gero, Whitehead & Rendell, 2016).

The production of the sperm whale click is a unique process: the animal produces a sound emission at the front of the head, in the “monkey lips”, which then bounces back into the head through the spermaceti. A small portion of the sound is emitted directly in the center (pulse p0), but most of it travels through the spermaceti towards the back of the head and reaches the frontal air sac, where the sound is reflected and returned forward, creating pulse p1. The reverberation is repeated several times, producing several pulses p2, p3, etc. Thus, the interval between impulses provides an indication of the duration required for the sound emission to travel through the head, and thus the size of the head. So the inter-pulse interval (IPI) has therefore become an important acoustic parameter that allows estimation of the size of individuals (Norris & Harvey, 1972; Gordon, 1991; Rendell & Whitehead, 2003; Rhinelander & Dawson, 2004; Growcott et al., 2011; Ferrari et al., 2024).

Several models have attempted to elucidate the spatial distribution patterns of sperm whales in their breeding grounds (Pirotta et al., 2011; Pace et al., 2018; Avila et al., 2022). The main physical habitat factors influencing the presence of sperm whales are thought to be water depth, followed by distance from the coast and slope of the seabed (Pace et al., 2018; Avila et al., 2022). However, according to Pace et al. (2018), the contribution of environmental variables depends on the type of sperm whale aggregation in the central Mediterranean. They found that the distribution of solitary whales was more explained by distance from the coast, and social units by slope (Pace et al., 2018). Sea surface temperature (SST) is thought to influence sperm whale distribution, with groups found in colder waters than solitary individuals are typically found in (Pirotta et al., 2011). This may be due to competition, where groups push solitary whales into warmer, less optimal waters, or a trade-off within groups that allows them to feed, interact, and care for young, unlike solitary individuals who focus solely on feeding. For most studies on sperm whale habitat, visual identification have been used to detect and locate sperm whales (Whitehead & Rendell, 2004; Praca et al., 2009; Pirotta et al., 2011; Pace et al., 2018). However, they characterised the sperm whales solely through visual observation, whereas acoustics, by analysing sperm whale clicks, can accurately determine the number of individuals present, their size (IPI) (Norris & Harvey, 1972; Gordon, 1991; Møhl et al., 2003; Rhinelander & Dawson, 2004; Growcott et al., 2011), and their behaviour (ICI) (Watkins & Schevill, 1977; Whitehead & Weilgart, 1990; Fais et al., 2015). In the Lesser Antilles, studies have been carried out to estimate the abundance (Vachon et al., 2022b) and health (Whitehead & Gero, 2015) of sperm whale populations, but also to understand their movement between different islands (Gero et al., 2007). A study of individual movement showed strong site fidelity in sperm whales, highlighting the importance of environmental parameters on culture at this spatial scale (Vachon et al., 2022a). On a smaller scale, the social structure of the sperm whale population in Guadeloupe and Dominica has been studied (Gero et al., 2014), as has their abundance (Gero & Whitehead, 2016; Rinaldi et al., 2021), revealing a decline in the latter. However, the sperm whale population in Martinique has never been characterised.

We combined visual and acoustic methods to characterise individuals by size (IPI calculation) and behaviour (observations and acoustic) in order to highlight intraspecific variations in sperm whale habitat use. The topographical factors analysed were bathymetry, distance from the coast and slope of the seabed. Social factors were the size of the aggregation, the average size of the individuals composing it and the presence of immatures which certainly affect the behaviour of females due to parental care. The time factor linked to changes in environmental parameters has also been added. In summary, the objective of this study was to characterise the intraspecific and spatial variation in habitat use of sperm whales in order to understand the factors that influence their distribution along the Caribbean coast of Martinique.

Materials and Methods

Field methods

For this study, researchers conducted boat expeditions along the west coast of Martinique to observe sperm whales and collect acoustic recordings. These excursions resulted in the development of a robust database to characterise the sperm whale habitat in this region. Along the west coast of Martinique, the seabed slopes steeply, rapidly forming an underwater canyon. This particular topography attracts sperm whales, which find it an ideal hunting ground for stalking their prey (Clarke, 1980). For data acquisition, the team adhered to and signed the respectful cetacean approach charter created by the AGOA Sanctuary (Sanctuaire, 2025).

One to two surveys per week were conducted from January 12 to May 15, 2024. A 6.5 m inflatable boat equipped with a 115 hp outboard motor was used for monitoring. The animals were located by acoustic and direct observations from the boat. Our study area extended from the southern part of the west coast of Martinique, along the Caribbean Sea side, at Cap Salomon (14° 30′ 27.878″N, 61° 6′ 2.787″W), to the northern part of the coast at Le Prêcheur (14° 48′ 6.139″N, 61° 13′ 27.636″W). For each day of the ship survey, the following weather conditions were noted: sea state (Beaufort scale), cloud cover (octa), wind direction and speed (knots), and visibility. We conducted our surveys from south to north, following the 1,500-m isobath, based on the location of historical observations. Ten acoustic points separated by 4 km were made along the 1,500-m isobath (Fig. 1). We did not make acoustic points, unless sperm whales had been heard and/or seen on the way to the port. The reasons we decided to follow this transect were (1) to maximise our chances of encountering sperm whales by following the isobath where sperm whales have often been sighted based on historical data, and (2) because sea conditions worsened with distance from the coast.

For each point of the transect, 2 to 5 min of recordings were made to try to detect animals. If the sperm whales were detected acoustically, we moved to locate them visually (depending on weather conditions and swell) and recorded all the individuals we detected visually. If no sperm whale was detected with the hydrophone, we proceeded to the next point. The sperm whales could be heard up to around 8 km, which means that they can be found up to the 2,000 m isobath. This detection range was estimated after tests at sea, which involved gradually moving away from a group of sperm whales sighted and assessing whether their clicks were still detectable by ear.

When sperm whales were sighted and approached at a minimum distance of 100 m, a GPS point was recorded and identification photos were taken using a Nikon D7100 camera with a 70–300 mm lens and a Canon Mark II 7D with a 70–200 mm lens. Then 5 to 10 min of continuous acoustic recordings were made. A single GPS point was recorded at the moment the sperm whales were sighted, and this same location was used as the reference point for both behavioural observations and acoustic recordings. From January to March, acoustic data were collected using an H2a-XLR omnidirectional hydrophone (Aquarian Audio Products; frequency response: 20 to 4.5 KHz, sensitivity: −180 dB re: 1V/ μPa) with 11 m cable connected to an iRig Pre 2 amplifier (frequency response: 50 Hz to 20 kHz) and a Zoom H1 handy recorder (sampling rate: 96 kHz, 24 bits). From April, an SQ26-08 omnidirectional hydrophone (Cetacean Research Technology; frequency response: 20 Hz to 50 kHz, sensitivity: −169 dB re: 1V/ μPa) replaced the Aquarian and the iRig Pre 2 amplifier and a Zoom H1n handy recorder (sampling rate: 96 kHz, 24 bits) replaced Zoom H1 handy recorder. Gain was adjusted between the two recorders to achieve approximately the same gain throughout the study. The Zoom H1 was set to 50/100, the iRig Pre 2 to 4/10 and the Zoom H1n to 7/10. From April onwards, a home-made satellite dish was used to make our listening more directional and to amplify the sound (Pavan, 2008). A JVC HA-S660 headset was connected to listen to the sounds in real time. Acoustic recordings were made in stereo at a depth of 10 m and stored in WAV format.

As soon as at least one individual from the aggregation had been seen to confirm the GPS point and a recording had been made, we continued on our way.

Additional data

We added cetacean observations from the Aquasearch historical database, collected from 28 April 2013 to 5 January 2024, while traversing the same study area, following the 1,000 to 1,500 m isobaths. Each row of the data table corresponded to a group of sperm whales observed, associated with a behaviour (resting, socialising or moving), its GPS position, the number of individuals, sex and age if possible, the presence of a calf and other details. We considered that the groups were equivalent to the aggregations in our study. The entire historical database is described in the study of Vries (2017).

The additional data did not include acoustic recordings. See Table 1 for details of additional data and Fig. 2 for details about sperm whale distribution from additional data. The nine additional observations for which GPS coordinates were not recorded were not included in the study. Three additional observations located slightly outside the study area were retained because acoustics at the edge of the area would have made it possible to detect sperm whales at these locations.

Table 1 Summary of sperm whale observations from additional datasets.

Source	Period	No. of observations	With behaviour data	With GPS data	
Aquasearch database	28/04/2013 to 05/01/2024	171	75	162	
Whale-watchers	12/01/2024 to 15/05/2024	19	2	19	

Figure 2 Oscillogram and spectrogram of the audio signal of clicks from two different individuals.

The oscillogram (top) shows the signal amplitude in the time domain. The spectrogram (bottom) shows the frequency energy distribution of the signal as a function of frequency (Hz) (overlap = 400, NFFT = 450). The decibel scale is added.

Habitat characterization

Each GPS point (historical and study data) was imported into QGIS v. 3.34.2 to obtain the following topographic information: bathymetry (in meters), distance to the coast (in meters) with an accuracy better than 100 m, and slope of the seabed (in percentage). Slope was calculated by dividing the height difference by the distance (about 500 m) around the observation point and was categorized as “light” (<10%), “medium” ([10–20%]) or “steep” ( ≥20%). The base maps used as well as the bathymetric lines were obtained from the available SHOM database (Service Hydrographique et Océanographique de la Marine (SHOM), 2025). For four GPS points, the slope value was not quantified because the individual was on an shelf. At the point under consideration, the slope was completely flat (zero), but on either side of the plateau, the inclination was in opposite directions. This meant that a single slope could not be defined at this point, making it impossible to calculate.

Aggregation definition

To standardise and facilitate the categorization of observed sperm whales, we defined a “aggregation” as a group of sperm whales observed within 1 km of each other that could be identified and distinguished acoustically. This distance was checked on QGIS during the analyses. In fact, sometimes several GPS points were taken in the field for different groups of sperm whales, which ended up belonging to the same aggregation (within 1 km). In addition, the signal had to be sufficient to allow calculation of the IPI. The term “aggregation” was used by Christal & Whitehead (1997), defining male individuals in temporal or spatial proximity to each other, heard within the 3–5 km range of the directional hydrophone. We reduced this distance after testing sperm whale detection with our hydrophone at such distances, and IPI measurement was impossible. See the next part for more details.

In order to characterise the aggregation with the topography, the following information was collected: GPS point (first animal seen), number of individuals seen on the surface, number of individuals heard, assumed age (adult, juvenile or immature), sex and all behaviours observed in the aggregation (hunting, socialising, moving and resting). All observed behaviours are described in Table 2 (the observation time of two to four scientists aboard the boat was at least 15 min). Observations complement acoustic recordings to determine certain behaviours. The presence of other species and the number of boats in the area were also recorded.

Table 2 Visual and acoustic description of the four behaviours identified.

The surface activity* ethogram was taken from Whitehead & Weilgart (1991). “Fluking” is in italics because it was only a clue to help identify the hunt. Acoustically it required the recording of regular clicks and buzzes.

Behaviour	Description		
	Acoustic	Visual	
Hunting	Presence of regular clicks and buzzes	Fluking*: whale raises its fluke above the water surface to an almost vertical position. Indicates the beginning of a foraging dive.	
Socialising	Presence of codas in recordings	Breach*: whale leaps partially or completely out of the water. Head-out*: whale raises head partially or completely above water surface. Lobtail*: whale thrashes fluke onto water surface. Side-fluke*: Whale turns on one side and partially lifts fluke out of the water. Fluke-first*: whale breaks the surface with the fluke first, frequently holding it in almost perfectly vertical position.	
Moving	none	The animal moves near the surface at a medium or fast speed, keeping the same direction, and does not dive.	
Resting	none	The animal remains in place or moves very slowly on the surface.	

In general, the number of individuals detected on acoustic recordings was greater than the number of sperm whales observed. Hence the interest in acoustics, which can be used to obtain data for species that remain underwater for long periods and at the surface for only a short time (Whitehead, 2003). For the same aggregation, several behaviours could be described. For example, acoustics often enabled us to define a hunting behaviour for individuals at depth, while an individual at the surface was seen resting.

Acoustic analyses

Sperm whale clicks were manually annotated using Audacity v. 3.4.2. To distinguish clicks from different individuals, click shape, ICI and IPI were taken into account (spectrogram parameters: fmin = 800 Hz; fmax = 10 kHz, window size = 256, window type = Hamming, gain = 20 dB, range = 70 dB).

Figure S1 shows a spectrogram and a 7-s waveform of a signal containing clicks from two individuals (A and B). Figure 2 shows an example of two clicks from two different individuals. A high pass filter was applied to all recordings. This filter reduced background noise from waves, boat activity, and isolated the signal above 1 kHz, where the energy of sperm whale clicks begins to emerge (Goold & Jones, 1995). This differentiation of clicks made it possible to count the number of individuals present in the aggregation (aggregation size).

To calculate the IPI, the waveform was used to detect the p1 and p2 amplitude peaks (p3 and p4 were very rarely visible on the signal). For each individual, the IPI was measured ten times on the recording and then averaged. Standard deviations were checked for errors (see Table S2 where standard deviations are given for Animal Size). To avoid errors due to water surface echoes, IPI were recorded at the beginning, middle and end of the recordings. Only IPI where the pulses were clearly visible were recorded. In Fig. 2 only the IPI of the second click could be calculated. As IPI can be falsely low due to clicks with a prolonged first impulse (p1), we limited the minimum IPI value to 2 ms (Marcoux, Whitehead & Rendell, 2006; Giorli & Goetz, 2020). This excluded possible clicks from the young immatures (Tønnesen et al., 2018). In our study, all observed immatures were acoustically characterised.

Animal size (AS) was then calculated from the IPI using the following two equations, the first for animals under 11 m (or IPI < 4.250 ms) (Gordon, 1991) Eq. (1) and the second for animals over 11 m (or IPI > 4.184 ms) (Growcott et al., 2011) Eq. (2):

(1) AS=4.833+1.453×IPI−0.001×IPI2

(2) AS=1.258×IPI+5.736

The method of using the two equations based on the IPI value was applied in two studies (Caruso et al., 2015; Poupard et al., 2022). Three classes of sperm whales have been identified. An IPI of less than 2.9 ms and an AS of less than 9 m corresponds to an immature whale, an IPI between 2.9 and 5.0 ms and an AS between 9 and 12 m corresponds to a juvenile and/or adult female, and finally an IPI greater than 5.0 ms and an AS greater than 12 m corresponds to an adult male (Gordon, 1991; Growcott et al., 2011).

Acoustic analyses revealed different types of aggregation, depending on the number of individuals identified and their average size within the aggregation, and the presence or absence of an immature.

Statistical analyses

All statistical analyses were performed with R software, version 4.2.2 (R Core Team, 2022). The alpha significance level was set to 0.05, indicating a 5% risk of rejecting the null hypothesis when it is true. For parametric tests, we always checked for normal distribution of model residuals by Shapiro-Wilk tests (Shapiro & Wilk, 1965), using the “shapiro_test” function from the rstatix package (Kassambara, 2022) and for homogeneity of variances by plotting the fitted values vs. the model residuals (Faraway, 2016). Details on the use of study data, acoustic and/or visual, and additional data have been added in Table S3.

The size of aggregations, the size of individuals and the presence of immatures are key factors in determining the social structures and behaviour of sperm whales (Best, 1979). Larger aggregations may reflect stable social units (Christal & Whitehead, 2001) or transient foraging associations formed for cooperative hunting (Kobayashi, Whitehead & Amano, 2020). The average size of individuals gives an indication of its composition and function-adult-dominated groups may consist of males seeking to reproduce, while aggregations of mixed size would indicate nursery units with immatures in need of parental care. The presence of juveniles also influences social cohesion, as guarding and protection behaviours strengthen social bonds and may be at the origin of specific behaviours and spatial distributions (Gero, Gordon & Whitehead, 2013). Taken together, these parameters provide a better understanding of sperm whale group dynamics and their ecological implications.

Social structure

First, the objective was to analyse whether the composition of aggregations (in terms of individual size) varied depending on the total number of individuals composing each aggregation. For each aggregation observed, the mean size of the individuals and the standard deviation of the sizes of the individuals present were calculated. Then the aggregations were grouped according to the total number of individuals they contained (e.g., all aggregations with two individuals together, those with three individuals together, etc). Finally, for each group of aggregations of the same size, the average individual sizes previously calculated were averaged. Similarly, the standard deviations of the individual sizes were averaged to obtain an overall measure of size variability in each aggregation category.

The influence of the presence of an immature (Nabsence = 8, Npresence = 11) on the aggregation size was tested using a Generalised Linear Model (GLM) with Poisson distribution, using the “glm” function. The absence of overdispersion was checked using the “check_overdispersion” function of the performance (Lüdecke et al., 2021) package to verify that the variance of the data was not too high. The P-value was calculated using a Type II Anova (McHugh, 2011) using the “Anova” function from the car package (Fox & Weisberg, 2019). The effect of the presence of an immature (Nabsence = 8, Npresence = 11) on the average size of the other individuals present was also tested using a Student’s T-test (Student, 1908), using the “lm” function. To do this, we calculated a new average of sizes by removing the sizes of immatures ( AS<9m).

Spatial distribution

In relation to topography, we tested the close association between bathymetry and distance from the coast (N = 27) using a Pearson correlation test (Freedman, Pisani & Purves, 2007), using the “cor.test” function. As we found a strong correlation between these two parameters, we decided to express only the bathymetric parameter in our analysis.

Variations in bathymetry and slope in relation to aggregation size (N = 19) and mean individual size (N = 19) were tested using Pearson correlation tests (Freedman, Pisani & Purves, 2007), using the “cor.test” function. These two parameters were then compared by class of individual (Nad.male = 28, Nad.female/juvenile = 30, Nimmature = 16) using linear models. P-values were calculated using Monte Carlo permutation tests with 1,000 resamples (Hothorn et al., 2008), using the “PermTest” function from the pgirmess package (Giraudoux, 2023). Finally, the bathymetry and slope distributions were tested according to the presence or absence of an immature (Nabsence = 8, Npresence = 11) using two Student’s T-tests (Student, 1908), using the “lm” function.

Habitat use

Several behaviours can be identified for an individual or an aggregation. Four observations without GPS point from historical data were not removed from the analysis. By adding historical data, we examined the distribution of behaviours according to bathymetry (Nhunting = 21, Nmoving = 71, Nresting = 12, Nsocialising = 10) using a linear model (“lm function”). Since multiple pairwise comparisons were conducted to compare bathymetry among behaviours, we applied the sequential Bonferroni correction to control for the increased risk of Type I errors (false positives) due to multiple testing (Abdi, 2010). This method adjusts the significance threshold for each comparison, reducing the likelihood of detecting spurious significant differences. P-values were calculated using Type II Anova (McHugh, 2011) using the “Anova” function from the car package (Fox & Weisberg, 2019). We also examined the distribution of behaviours according to slope (Nhunting = 20, Nmoving = 70, Nresting = 12, Nsocialising = 10) using a linear model (“lm function”). The P-value was calculated using a Monte Carlo permutation test with 1,000 resamples (Hothorn et al., 2008), using the “PermTest” function from the pgirmess package (Giraudoux, 2023). See Table S4 for details on sample sizes for each behaviour, separating the additional data from the data from our study.

Temporal variation

The aim of the temporal and spatial analysis of sperm whales is to gain a better understanding of sperm whale population dynamics, in relation to weather conditions and prey distribution. Meteorological factors such as humidity and temperature can affect ecological dynamics, for example by influencing the availability of resources (Rosa et al., 2017; Guerra et al., 2011), and therefore indirectly animal movements. In order to visualize the temporal variations in the structure of aggregations (AS and aggregation size analyses), we have grouped the months from January to March into a single period (period 1), corresponding to the dry season in Martinique, and the months from April to May into a second period (period 2), characterised by a wetter climate. This second period also coincides with the gradual departure of the males and the end of the breeding season (B de Montgolfier, 2020, personal communication).

Temporal and spatial distribution in relation to bathymetry (number of aggregations, NJanuary = 43, NFebruary = 30, NMarch = 30, NApril = 22, NMay = 9) and in relation to the slope (NJanuary = 43, NFebruary = 30, NMarch = 28, NApril = 22, NMay = 9) were tested using linear models with permutation tests (1,000 permutations), using the “PermTest” function from the pgirmess package (Giraudoux, 2023). For these tests, we used the study data and additional data corresponding to the months of the study period. At the level of aggregation, we ran a GLM with Poisson distribution to see if the size of aggregations varied over the months (N = 24), using the “glm” function. We checked the absence of overdispersion using the function “check_overdispersion” from the package performance (Lüdecke et al., 2021). The P-value was calculated using a Type II Anova (McHugh, 2011) using the “Anova” function from the car package (Fox & Weisberg, 2019). Finally, we compared changes in the size of the individuals found over time (number of individuals, Nperiod1 = 38, Nperiod2 = 36) using a linear model. The P-value was calculated using a Type II Anova (McHugh, 2011) using the “Anova” function from the car package (Fox & Weisberg, 2019). For these last two analyses, only the data from this study were used.

Results

From 12 January to 15 May 2024, 24 days of data collection were conducted, covering a total distance of 1,986 km. The two cameras captured 984 sperm whale images. A total of 19 h 45 min of recordings were made. Figure 3 illustrates the distribution of sperm whales from 2013, highlighting a concentrated distribution between the 1,000 and 2,000 m isobaths. The description of the number of behaviours characterised visually and acoustically for our study data is shown in Table S4. In total, 19 aggregations of sperm whales were acoustically characterised and 74 IPIs were calculated.

Figure 3 Sperm whale observations distinguishing between additional data from 28 April 2013 to 26 March 2024 (including observations from the whale-watcher and study data from 12 January to 15 May 2024).

A total of 27 aggregations were found, but GPS coordinates were not taken for one of them.

Social structure

Sperm whales were observed in 42% of the 24 boat surveys, with a total of 19 aggregations and 74 individuals recorded and acoustically characterised (Fig. 4). The number of aggregations characterised per day ranged from one to four and the number of individuals from one to 16. Seven observations were not associated with acoustic data, due to poor quality recordings. Of the 19 aggregations characterised acoustically, 15 contained adult males (79%), 14 adult females or juvenile males (74%) and 11 immatures (58%). Surprisingly, two aggregations included only adult and immature males, with no females identified. Details of the composition of the classes of individuals in each aggregation are given in Table S1.

Figure 4 By release date of ship surveys, number of aggregations and total number of acoustically identified individuals.

The absence of a bar means that no sperm whales were observed or listened to.

Aggregation size varied from 1 to 9 (median = 4, Q1 = 2, Q3 = 5) and individual IPI from 1.90 to 7.80 ms (median = 3.85 ms, Q1 = 2.90 ms, Q3 = 5.38 ms, x¯ = 4.18 ms), corresponding to an AS of 7.6 to 15.6 m (median = 10.5 m, Q1 = 9.0 m, Q3 = 12.5 m, x¯ = 10.8 m). An IPI of 1.90 ms was added despite its value being less than 2 ms because the presence of the calf was confirmed visually. The distribution of AS is illustrated in Fig. 5 and more details is reported in Table S2. During the data collection period, immatures (N = 16), females/juvenile males (N = 31) and adult males (N = 27) were present in the study area. As immatures were observed during the study, the presence of adult females and immature individuals suggests parental care.

Figure 5 Distribution of animal sizes, highlighting the three sperm whale classes (Nad.male = 27, Nad.female/juv.male = 31, Nimmature = 16).

Regardless to the size of the aggregations, the average size of the individuals composing them showed little variation, ranging from 9.3 to 12.4 m (Fig. 6). However, as aggregation size increased, the difference in sizes among individuals within these groups also increased from 1.3 to 3.7 m. This pattern highlights the presence of individuals of all age classes in large aggregations, including immatures, juveniles, and adult males and females. However, for aggregations of seven or nine individuals, the standard deviation was low.

Figure 6 Mean and standard deviation of animal size by aggregation size.

When an immature individual was present, aggregations were larger in size and composed of smaller individuals on average (Table 3, Figs. 7A, 7B).

Table 3 Statistical results for aggregation size and mean individual size in the presence of an immature individual.

Significance levels are indicated as follows: * for P-values between 0.005 and 0.01, ** for P-values < 0.01, and *** for P-values < 0.001.

Variable	Test	Statistic	P-value	Interpretation	
Aggregation size	Overdispersion	Dispersion ratio = 0.813, Pearson’s χ2=13.824	P=0.679	No overdispersion	
	GLM	χ2=3.815	P=0.041	*	
Mean individual size	Shapiro-Wilk	W=0.970	P=0.781	Normal distribution	
	T-test	t=2.115, df=17	P=0.049	*	

Figure 7 Aggregation size (A) and average size of individuals in the aggregation (B) according to the presence or absence of immature(s).

Medians and standard deviations are represented.

Spatial distribution

As expected, there is a significant positive correlation between bathymetry and distance from the coast (Table 4). The spatial distribution of sperm whales was analysed based on parameters: aggregation size, average individual size, their class (immature, adulte female or juvenile male, adulte male) and the presence of immatures. All these parameters were tested in relation to bathymetry and slope. The statistical results for this part are given in the Table 4 and the parameters did not vary with bathymetry or slope.

Table 4 Statistical results showing the influence of different parameters on the spatial distribution of sperm whales.

Independent variable	Dependent variable	Test	Statistic	P-value	Interpretation	
Distance from the coast	Bathymetry	Shapiro-Wilk	W = 0.977	P=0.801>0.050	Normal distribution	
	(Fig. S2)	Pearson correlation	β=0.134±0.012,SE,	P<0.001	Strong positive correlation	
			t=11.640, df=25,			
			R2=0.844			
Aggregation size	Bathymetry	Shapiro-Wilk	W = 0.954	P=0.436>0.050	Normal distribution	
	(Fig. S3a)	Pearson correlation	t=0.029,	P=0.977	No significant correlation	
			df=17			
	Slope	Shapiro-Wilk	W=0.940	P=0.268>0.050	Normal distribution	
	(Fig. S3b)	Pearson correlation	t=−1.218,	P=0.240	No significant correlation	
			df=17			
Average size of individuals	Bathymetry	Shapiro-Wilk	W=0.953	P=0.444>0.050	Normal distribution	
	(Fig. S3c)	Pearson correlation	t=0.167,	P=0.870	No significant correlation	
			df=17			
	Slope	Shapiro-Wilk	W=0.914	P=0.086>0.050	Normal distribution	
	(Fig. S3d)	Pearson correlation	t=0.071,	P=0.944	No significant correlation	
			df=17			
Sperm whale classes	Bathymetry	Linear model (perm. test)		P=0.474	No significant difference	
	Slope	Linear model (perm. test)		P=0.210	No significant difference	
Immature presence	Bathymetry	Shapiro-Wilk	W=0.953	P=0.445>0.050	Normal distribution	
	(Fig. S3e)	T-test	t=−0.437,	P=0.668	No significant difference	
			df=17			
	Slope	Shapiro-Wilk	W=0.904	P=0.058>0.050	Normal distribution	
	(Fig. S3f)	T-test	t=−0.905,	P=0.378	No significant difference	
			df=17			

Habitat use

We observed significant differences in the execution of behaviours related to bathymetry (Shapiro-Wilk test: W=0.984, P=0.202; LM: F=7.479, df=3, P<0.001) (Fig. 8A). Post-hoc comparisons showed that sperm whales were hunting at deeper bathymetries (median = 1,700 m, x¯ = 1,643 m) more than when moving (median = 1,300 m, x¯ = 1,320 m) or socialising (median = 1,200 m, x¯ = 1,180 m), but there were no differences with resting behaviour (median = 1,400 m, x¯ = 1,550 m). Bathymetry for the other three behaviours was not significantly different (see Table 5 for post-hoc results). However, there was no significant difference between the behaviours with respect to slope (LM with permutation test: P=0.390) (Fig. 8B).

Figure 8 Bathymetry (A) and slope of the seabed (B) according to the different behaviours.

Medians and standard deviations are represented. Different letters indicate significant differences and identical letters indicate no significant difference.

Table 5 Confusion matrix showing different behaviours.

Results of pairwise comparisons of bathymetry after sequential Bonferroni-Holm correction. Bold text indicates a significant difference, roman text indicates no difference.

	Hunting	Moving	Resting	Socialising	
Hunting					
Moving	β = −323.14 ± 78.39 SE				
	F-value = 16.991				
	t-value = −4.122				
	P < 0.001				
Resting	β = −92.86 ± 143.30 SE	β = 230.28 ± 101.48 SE			
	F-value = 0.420	F-value = 5.150			
	t-value = −0.648	t-value = 2.269			
	P = 0.522	P = 0.026			
Socialising	β = −462.90 ± 131.70 SE	β = −139.70 ± 101.30 SE	β = −370.00 ± 166.00 SE		
	F-value = 12.352	F-value = 1.902	F-value = 4.968		
	t-value = −3.514	t-value = −1.379	t-value = −2.229		
	P < 0.001	P = 0.172	P = 0.037		

Temporal variation

Between the two periods, AS significantly decreased (see Table 6, Fig. 9). Regarding social structure, the aggregation size remained unchanged between over the months (see Table 6 and Fig. S4c).

Table 6 Statistical results for behaviours related to bathymetry, slope, and social structure across two periods.

Significance levels are indicated as follows: * for P-values between 0.005 and 0.01, ** for P-values < 0.01, and *** for P-values < 0.001.

Variable	Test	Statistic	P-value	Interpretation	
Animal size	Shapiro-Wilk	W=0.975	P = 0.147	Normal distribution	
	T-test	βperiod2=−1.526±0.459	P = 0.001	***	
		t=3.325			
		df=72			
Aggregation size	Overdispersion	Dispersion ratio = 1.052	P = 0.394	No overdispersion	
		Pearson’s χ2=16.936			
	GLM	χ2=0.995	P = 0.318	N.S.	
Bathymetry	LM with permutation test		P = 0.067	N.S.	
Slope	LM with permutation test		P = 0.279	N.S.	

Figure 9 AS between the two periods. Medians and standard deviations are represented.

Otherwise, there was no change in the distribution with respect to bathymetry or slope (Table 6 and Figs. S4a and S4b).

Discussion

The study area corresponds to a sperm whale breeding area (Gero et al., 2014). According to our results, the structure of the aggregations was consistent with the literature, as females, juveniles and immatures were found forming social units (Whitehead, 2003), often accompanied by mature males approaching from the north to breed (Gero et al., 2014). The temporal analysis showed a change in the structure of the aggregations between the beginning and the end of the study period, suggesting the departure of males to the north. As for the spatial distribution of the population characterised during the study, it remained constant according to the bathymetry and the slope of the bottom, regardless of the type of aggregation and the classes of sperm whales. However, sperm whales showed a spatial variation in habitat use in relation to bathymetry.

Comparison of social structure with other regions

Of the 19 identified aggregations, only one contained a single individual, an adult male hunting along bathymetric depths up to 1,500 m of bathymetry. All others contained at least two individuals (see Table S1). As the size of the aggregations increases, the standard deviation of sizes between individuals also seems to increase, although the means remain fairly similar, indicating the presence of individuals with more size differences (immature–juvenile and/or adult female–adult male) in the larger aggregations. On the other hand, the standard deviation remains low for aggregations of seven and nine individuals. This highlights the fact that the structural dynamics of aggregations are not totally defined by the number of individuals in them. Furthermore, when an immature was present, the aggregation was larger and nearby individuals were significantly smaller. The social structure found in our study area was consistent with the literature, as females, juveniles and immatures form stable social units in tropical and subtropical waters (Whitehead, 2003). For example, in Dominica, aggregations encountered averaged 7–9 individuals, with the majority of social units containing females and juveniles (Gero et al., 2014). Within identified groups of more than four individuals, there was almost always at least one male. Again, in Dominica, mature males were identified in close proximity to social units, and groups were larger when a mature male was present (Gero et al., 2014). In northern Chile, mature males tend to accompany some groups of females (Coakes & Whitehead, 2004). The presence of mature males in nearly 80% of the characterised aggregations was not surprising, as they come to breed in the Caribbean (Gero et al., 2007; Whitehead, 2018).

This dynamic highlights how the composition of sperm whale aggregations can vary across regions. Indeed, the proportions of juvenile males/females, adult females, adult males, and juveniles within these groups have been found to differ between studies. For example, the percentage of adult males in our study area was higher than in a study from the Ionian Sea (Caruso et al., 2015) but similar to another study from the Mediterranean Sea (Poupard et al., 2022). Jaquet (2006) showed that the Gulf of Mexico population (median = 9.3 m) was composed of smaller animals than the Gulf of California population (median = 10.7 m) thanks to photogrammetric measurements. Only a few of adult males were found in both areas (Jaquet, 2006). Although the methods are different with photogrammetry which allows the measurement of sizes less than 7.7 m (IPI< 2 ms) (Jaquet, 2006), these comparisons highlight the variability in the size distributions depending on the geographical areas, even close ones (Caruso et al., 2015; Poupard et al., 2022) and certainly also on the duration of the study, in connection with the migrations of the males (Dufault, Whitehead & Dillon, 1999; Whitehead, 2003).

What was surprising was the detection of males with immatures and no females nearby for two of the aggregations defined. Several hypotheses are possible, the first being the presence of females in the area but vocally inactive. Moreover, we recognise that there may be errors in the number of individuals per aggregation since it is defined by acoustics, and sperm whales can be silent. The absence of clicks may indicate that they are resting and not hunting or socialising (Watkins, 1980). A second hypothesis is that the immature males detected are close to maturity and are beginning to interact with other males. Although the males do not seem to form lasting or privileged bonds with each other, groups of males have already been seen, certainly linked to environmental factors such as the presence of prey (Whitehead, Brennan & Grover, 1992; Lettevall et al., 2002).

Spatial distribution independent of aggregation type

Contrary to the model developed by Pace et al. (2018) highlighting a variation in spatial distribution according to the social composition of the groups, this was not the case in our study. Indeed, we did not observe any variation in bathymetry as a function of aggregation characteristics (such as size of aggregation, average size of individuals, or presence of immatures). Given that sperm whales belong to the same population and social units, and may even contain the same individuals on different days (Dufault, Whitehead & Dillon, 1999; Whitehead, 2003), it is possible that they share similar preferences in their spatial distribution. An alternative hypothesis is that variations in bathymetry was not strong enough to induce changes in spatial distribution between aggregation types. However, since immatures need to nurse and these events occur at depths below 30 m (Sarano et al., 2023), one would have expected the presence of immatures to influence the distribution as a function of bathymetry. One hypothesis that could explain the lack of variation is that alloparenting between females within social units (Whitehead, 1996) may well compensate for the need to surface frequently by remaining in the most productive areas. Furthermore, we did not distinguish calves from other immatures that no longer nurse. Similarly, distribution did not differ by sperm whale class or by the average size of individuals within an aggregation. This finding was unexpected, as it is generally understood that larger individuals tend to dive deeper due to their greater breath-holding capacity (Drouot, Gannier & Goold, 2004). This distinction may not be visible in our study area, because it mainly characterises mature males, which are more offshore in areas with deeper bathymetry (Drouot, Gannier & Goold, 2004). Here, the priority for males would be to reproduce (Dufault, Whitehead & Dillon, 1999; Whitehead, 2003) and therefore be close to females.

With regard to slope, no significant variation was observed in the distribution of sperm whales. The data indicate a marked homogeneity of the seabed within the areas frequented by these cetaceans, suggesting that slope is not a determining factor in their spatial distribution on a small scale.

Specialised areas of behaviour

Although all aggregations were distributed similarly, sperm whale behaviour varied with bathymetry. Specifically, a bathymetric area appeared to be preferred for hunting ( x¯ = 1,643 m), distinct from the areas used for movement ( x¯ = 1,320 m) and socialisation ( x¯ = 1,180 m) but comparable to those used for resting ( x¯ = 1,550 m).

In contrast to social behaviour, hunting occurs at greater depths, likely along submarine canyons where giant squid (Dosidicus gigas), a preferred prey of sperm whales (Clarke, 1980), are found (Martinique Fisheries community, personal communication). The deeper bathymetries observed in hunting areas likely reflect the depth preferences of these mesopelagic cephalopods (Nigmatullin, Nesis & Arkhipkin, 2001). Furthermore, the similarity in bathymetries between resting behaviours ( x¯ = 1,550 m) and hunting behaviours ( x¯ = 1,641 m) may indicate a need for rest after deep, high-energy dives (Watwood et al., 2006).

Socialising behaviour was expected at shallower depths, as groups with immature were anticipated to socialise more in these areas, due to lactating which takes place rather on the surface (Sarano et al., 2023). However, as no influence of aggregation size, average individual size, or the presence of immature individuals on spatial distribution was found, we suggest that behaviours are not influenced by the type of the aggregation or that there are more important factors, such as prey availability (Whitehead & Weilgart, 1991; Watwood et al., 2006) or environmental conditions (Jaquet & Whitehead, 1996; Pirotta et al., 2011; Wong & Whitehead, 2014) that explain the identified behaviours. Moreover, in areas of high maritime traffic, social cohesion may help cetaceans cope with noise. For example, an increase in social signals has been observed in response to sonar noise in the northern Norwegian Sea (Curé et al., 2016). In other cetaceans, such as blue whales (Balænoptera musculus), vocalizations increase during social interactions in the presence of high seismic activity (Di Iorio & Clark, 2010). Among odontocetes, bottlenose dolphins (Tursiops truncatus) have been observed to increase their whistling during the early stages of a jet-ski approach, possibly reflecting an increased need for cohesion (Buckstaff, 2004). Furthermore, groups containing mother-calf pairs whistle more in the presence of fast-moving boats compared with groups without calves (Guerra et al., 2014).

As for spatial distribution, behaviour did not change as a function of slope, suggesting that the variations in slope are not sufficiently marked to cause a change in their spatial distribution. Unlike other cetacean species—such as humpback whales, which migrate to Caribbean waters to reproduce and give birth while fasting (Martin et al., 1984), sperm whales do not follow this pattern and feed during their breeding season.

Finally, we can conclude that there was no intraspecific spatial segregation according to topography within our study area. This means that even if certain areas are preferred for hunting or socialising, all groups, regardless of individual size, aggregation size or the presence of juveniles, use similar depths, distances from the coast and slopes for these activities. Their habitat selection seems to be mainly determined by the bathymetry rather than by local gradients in the ocean floor (Pace et al., 2018).

As the historical data did not include acoustic information, hunting behaviours were not recorded and socialisation behaviours (N = 10) were probably underestimated due to the lack of detection of codas. Moreover, even sperm whales are great divers, reaching depths of well over 1,500 m (Teloni et al., 2008). As we move along the 1,500 m bathymetry line, our visual range is restricted: we may miss a sperm whale that would be on the 2,000 m isobath, for example, if it doesn’t click. Of course, we can also miss sperm whales that emit vocalizations at distances greater than 8 km. Furthermore, it is essential to recognise that the additional data was reported by a variety of people and that the definition of behaviour was not always the same, since observations always reveal a degree of subjectivity.

Temporal variation

A temporal variation between January-March and April-May was observed. Indeed, AS decreased between the two periods ( x¯period1 = 11.6 m, x¯period2 = 10.1 m). As the size of individuals decreased but the number of individuals per aggregation did not vary over the months, aggregations may have consisted of more females/juvenile males and immatures compared to males. Both results suggest that males began to move out of the breeding area to higher latitudes (Dufault, Whitehead & Dillon, 1999; Whitehead, 2003). However, a full-year study would confirm this hypothesis and improve our knowledge of the structural and temporal dynamics of Caribbean sperm whales.

The sperm whales did not show any temporal pattern in their spatial distribution in relation to the topography. In fact, from January to May, the distribution according to bathymetry and slope did not vary, suggesting a homogeneous habitat and little ecological variation in the area.

Influence of other factors

Although the spatial distribution of sperm whales was studied, other environmental factors such as SST (Pirotta et al., 2011; Wong & Whitehead, 2014), seabed characteristics, seamounts, slope orientation (Quetglas, Carbonell & Sánchez, 2000; Pirotta et al., 2011) and wind strength (Jaquet & Whitehead, 1996) could play a critical role in biomass distribution (Jaquet & Whitehead, 1996; Wong & Whitehead, 2014) at spatial scales larger than that of our study area, of the order of several thousand km2. As sperm whales spend about three-quarters of their time searching for prey (Whitehead & Weilgart, 1991; Watwood et al., 2006), their distribution depends on the distribution of cephalopods (Clarke, 1980; Wong & Whitehead, 2014), but also fish, which are an occasional food source for sperm whales (Clarke, Martins & Pascoe, 1993). Pirotta et al. (2011) highlighted the link between the distribution of sperm whales at bathymetries deeper than 2,000 m and the presence of cephalopods in the Balearic region (Pirotta et al., 2011). Unlike our study, sperm whale habitat modeling studies have focused on larger geographic areas and longer time periods. For example, Pirotta et al. (2011) studied the distribution of sperm whales around Ibiza, Mallorca and Menorca for 6 years, where they found temporally stable preferential areas, areas where no sperm whales were observed, and a preference for warmer areas. As for Vachon et al. (2022a), data were collected in the Eastern Caribbean over a period of more than a year and approximately 700 km, allowing the examination of geomorphic features such as canyons, plateaus, escarpments, slopes and abyssal plains. Their result over a period of more than a year showed that the distribution of sperm whales depended more on site fidelity, rather than environmental parameters, is the main driver of sperm whale distribution (Vachon et al., 2022a). In addition to environmental factors, the presence of other animals, such as competitors or predators, may also influence the distribution and habitat use of sperm whales. During our observations, we observed harassment on two occasions, first by pilot whales (Globicephala macrorhynchus) and then by killer whales (Orcinus orca). These were the only two times that codas were recorded during our study. The aggressive behaviour of pilot whales was previously documented by Díaz-Gamboa, Gendron & Guerrero-de la Rosa (2022) in the Gulf of California. Their biopsies showed that both species preferred to feed on giant squid (Díaz-Gamboa, Gendron & Guerrero-de la Rosa, 2022), indicating that they are competitors for the same food source. On the other hand, killer whales have been shown to be predators of sperm whales (Pitman et al., 2001), which interrupt their foraging or resting dives when they hear killer whale sounds and return to the surface, initiating a significant degree of social behaviour (Curé et al., 2013).

Choice of methodology

In terms of methodology, several approaches can be used to calculate the IPI, including the manual method applied in this study. According to Antunes, Rendell & Gordon (2010), the manual method, while time-consuming, is considered more accurate and less sensitive to noise than automatic detection. However, automatic methods offer significant advantages, particularly in processing large datasets, despite their reliance on a high signal-to-noise ratio and the orientation of the animal relative to the hydrophone, which can lead to false negatives (Antunes, Rendell & Gordon, 2010). Over the years, automatic methods have demonstrated their effectiveness, as suggested by the comparable size estimates obtained using IPI calculations and photogrammetry (Rhinelander & Dawson, 2004; Growcott et al., 2011). Additionally, a comparison of three automatic methods (envelope, cepstrum, and cross-correlation) highlighted variations in their efficiency, with the envelope method providing the most accurate estimates (Bøttcher et al., 2018). While both approaches have their strengths and limitations, the continuous development of automatic methods remains crucial for handling large volumes of data while aiming to minimize subjectivity in manual annotation. It is also important to note that biases may exist in the conversion of IPI to individual size. Each equation (Gordon, 1991; Growcott et al., 2011), has its own potential margin of error and is based on specific populations.

This subjectivity is also reflected in the lack of standardisation of terms used to describe cetacean structures, which makes comparisons between different studies problematic. Due to the difficulty in accurately determining the distance of individuals from the hydrophone, it was not possible to assign a specific name to the aggregation found in the literature. For example, a group can be defined as a set of individuals moving together in a coordinated manner for periods ranging from a few hours to a few days, while a cluster would rather be like a transient subset of a group consisting of individuals swimming side by side in a coordinated manner (Cantor & Whitehead, 2015). In contrast, other authors have defined a group as about twenty individuals feeding in structured formations over about 1 km for days, while a cluster consisted of at least two individuals grouped together at the surface for about 10 min (Lettevall et al., 2002). According to these definitions, our term “aggregation” could correspond to “group,” but no two studies used the same methodology as ours. These differences in definition highlight the problem of standardising terminology and methods used in sperm whale studies, making comparisons between studies difficult.

The inclusion of additional data in our study is a crucial resource for analysing the distribution of behaviour over time. Indeed, socialisation behaviour, for example, remains exceptionally rare in our dataset (Ntotal = 10, N = 3 in our study). While historical recordings provide valuable information, feeding behaviour could not be explicitly defined, as its identification depends on acoustic analysis—an aspect that has not been achieved in previous studies. The absence of acoustic recordings also prevents us from determining key ecological parameters, such as the size of aggregations and the average size of individuals. This gap highlights the need to incorporate bioacoustic monitoring into future research in order to refine behavioural classifications and improve our understanding of population dynamics. Furthermore, even though the additional data were integrated with those from this study, the observation duration per sperm whale group was not systematically recorded, limiting our ability to assess encounter precision. It was estimated that the time spent near groups of sperm whales to collect data was certainly similar, as sperm whales stay at depth for a long time, so the time dedicated to observation is short, and regulations limit the amount of time we are allowed to spend near a marine mammal.

Conservation and management implications

Our study provides valuable information on the use of sperm whale habitat along the west coast of Martinique, highlighting the key environmental and social factors shaping their distribution. This was made possible by focusing on a fine spatial scale and a complementary methodology, combining visual and acoustic data. This fine-scale approach establishes a robust foundation for implementing effective conservation and spatial management strategies. As MarineTraffic’s year-on-year data shows, the risk is along the Caribbean coast, not in the Atlantic, where the near-shore topography is not consistent with sperm whale habitat and where shipping traffic is much lower.

One of the main external factors influencing cetacean behaviour is maritime traffic (Erbe et al., 2016). The Caribbean Sea, including the waters around Martinique, is heavily trafficked (Fig. 1), posing two main threats to sperm whales: ship strikes and noise pollution. Ship strikes are one of the main causes of mortality for large cetaceans (Peltier et al., 2019). Although the implementation of measures such as speed limits (<5 kt) and a regulated distance between the ship and the animal of more than 300 m within the Agoa sanctuary helps mitigate these dangers, the knowledge of the population is essential to understand all the issues and improve the regulations in the future.

In addition to the physical obstacle represented by ships, they emit continuous noise that can mask the sounds of marine mammals (Erbe et al., 2016). Sperm whales use echolocation to feed, navigate, and communicate (Whitehead & Weilgart, 1991; Watwood et al., 2006). Thus, increasing underwater noise from navigation, but also from seismic surveys and industrial activities interferes with these essential behaviours through acoustic masking (Erbe et al., 2016). Chronic exposure to noise can lead to habitat displacement, reduced foraging efficiency, and increased stress levels (Wright et al., 2007), phenomena that can also occur in the Martinique sperm whale population. Passive acoustic monitoring could provide more information on how noise pollution affects this species (Poupard et al., 2022) and help develop targeted mitigation measures.

Given these external pressures, our results may contribute to further conservation efforts. Identifying and strengthening the protection of key habitat areas for sperm whales, particularly those used for socialisation and foraging, could reduce the risk of disturbance. In addition, improved real-time whale tracking systems, such as Repcet® (Campana et al., 2015), could help reduce the risk of collision by alerting vessels to the presence of cetaceans. Indeed, it would be interesting to update the software for the west coast of Martinique to warn sailors that sperm whales are mainly found south of Saint-Pierre at depths of over 1,000 m, that they are often far apart and almost never alone, and in particular that there are always individuals at depth when some are on the surface. In particular, more attention should be paid when a calf is present as the group may be larger.

Combining long-term passive acoustic monitoring with vessel tracking data could offer a more comprehensive understanding of how human activities influence the ecology of sperm whales. Closer collaboration between researchers, conservation organizations, and maritime authorities will be essential to ensure the coexistence of marine megafauna and sustainable human activities in the Caribbean Sea.

Conclusions

Our study, based on 24 vessel surveys over 5 months, represents the first comprehensive analysis of sperm whale habitat use in Martinique waters.

The distribution of aggregations along the Martinique coastline will make it possible to develop targeted alerts or recommendations to reduce collisions between ships and sperm whales, particularly via the Repcet® system developed in France. This system allows vessels to communicate information about the presence of cetaceans at the surface. When a cetacean is spotted, a detection radius expands over time. This radius is calculated based on a single individual, but our study shows that sperm whales are frequently observed in groups in Martinique. Therefore, the detection radius in this area should be increased to better reflect reality. In the future, sustained long-term studies are essential to capture seasonal variations in the spatial distribution of sperm whales along the Caribbean coast of Martinique. Continuous monitoring will refine our understanding of their movement dynamics and guide adaptive management strategies.

Supplemental Information

Supplemental Information 1 Sperm whale Recording made on 01/12/2024.

Supplemental Information 2 Sperm whale recording made on 01/19/2024.

Supplemental Information 3 Sperm whale recording made on 01/26/2024.

Supplemental Information 4 Sperm whale recording made on 01/29/2024.

Supplemental Information 5 Sperm whale recording made on 02/01/2024.

Supplemental Information 6 Sperm whale recording made on 02/22/2024.

Supplemental Information 7 Sperm whale recording made on 02/29/2024.

Supplemental Information 8 Sperm whale recording made on 03/21/2024.

Supplemental Information 9 Sperm whale recording made on 04/29/2024.

Supplemental Information 10 Sperm whale recording made on 05/01/2024.

Supplemental Information 11 Data.

Supplemental Information 12 Supplementary material.

Thank you to the whale-watcher Yannick Tursi from Planète Dauphin and the whole Aquasearch team for participating in the data collection.

Additional Information and Declarations

Competing Interests

The authors declare that they have no competing interests.

Author Contributions

Séréna Laurent conceived and designed the experiments, performed the experiments, analyzed the data, prepared figures and/or tables, authored or reviewed drafts of the article, and approved the final draft.

Marion Poupard conceived and designed the experiments, performed the experiments, analyzed the data, prepared figures and/or tables, authored or reviewed drafts of the article, and approved the final draft.

Célia Ortolé performed the experiments, authored or reviewed drafts of the article, and approved the final draft.

Céline Valin performed the experiments, authored or reviewed drafts of the article, and approved the final draft.

Benjamin de Montgolfier conceived and designed the experiments, performed the experiments, analyzed the data, authored or reviewed drafts of the article, and approved the final draft.

Data Availability

The following information was supplied regarding data availability:

The raw data are available in the Supplemental Files.

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
