# Peer review of "Intraspecific and spatial variation in habitat use by sperm whales (Physeter macrocephalus) along the west coast of Martinique"

_PeerJ, doi:10.7717/peerj.19614_

## Round 0.1 · original submission · Minor Revisions

Dear Dr. De Montgolfier,

After receiving three reviews of your work, they are all pleased with it. Nevertheless, some questions need to be addressed, especially regarding the inclusion of historical records and some definitions for wider audiences. Please pay attention to these aspects when preparing the new version of your manuscript and your rebuttal letter.

Reviewer 1 ·

Basic reporting

This study investigates intraspecific and spatial variation in habitat use by sperm whales (Physeter macrocephalus) along the west coast of Martinique. Using visual and acoustic data collected across 24 surveys, the research characterizes individual size, behavior, and social structures in relation to habitat features such as bathymetry and proximity to shore, providing valuable insights for conservation and maritime traffic management strategies.
The manuscript is written in clear, professional English, but there are areas where clarity and fluency could be improved. Minor grammatical errors and redundancies should be addressed to enhance readability. My comments and suggestions are provided in the attached document.
The authors are encouraged to integrate their results into a broader ecological framework and to discuss external factors such as maritime traffic and anthropogenic noise in more detail. This would strengthen the implications of their findings for habitat conservation and spatial management.

Experimental design

The experimental design is robust and appropriate for addressing the research question. With minor clarifications regarding methodological choices and data integration, the study would meet the standards for reproducibility and technical rigor.

Validity of the findings

The findings are valid, well-supported by robust data and rigorous analyses, and the conclusions are appropriately framed within the scope of the results.

Additional comments

no comment

Annotated reviews are not available for download in order to protect the identity of reviewers who chose to remain anonymous.

Reviewer 2 ·

Basic reporting

The manuscript is well-written overall, but there are areas requiring attention to clarity, grammar, and structure.
Key Issues
1. Repcet (Real-Time Cetacean Tracking) (Lines 52-53): This software is mentioned a few times throughout the text, the first time being in the introduction, but I don’t see the relevance. I would suggest elaborating on how your findings would be integrated into the software, it would make more sense mentioning it.
2. Clarity in identifying the novelty (Lines 44–49): The manuscript states that the sperm whale population in Martinique has not been previously characterized but does not provide sufficient context about why this is significant. I would add more detail about how these findings address broader ecological or conservation challenges specific to the region.
3. Grammar and clarity issues:
o Lines 58-59: “Thanks to their spermaceti organ..” is very causal and should be revised.
o Line 61: I’ve never heard the reference “two distinct lifestyles”, I would suggest revising to maintain consistent with existing literature.
o Line 56 and 68: The word odontocete should not be capitalized.
o Lines 77-78: Inter-Pulse interval should be defined briefly in this sentence before you go into the details.
o Lines 98-101: I think you mean to say visual identification has “…been used to detect and locate sperm whales” instead of passive acoustics
o Line 120: "Field methods" should introduce a clear, concise overview of the methodology before diving into details.
o Lines 203-204: This sentence doesn’t make sense. What does (2 to 5) mean?
o Line 277: I’m not sure you say ‘higher bathymetries’, perhaps ‘deeper bathymetries’.
o Lines 311-321: This paragraph lacks focus and organization. I would suggest strengthening the connections to these other studies with the results in your study.
o Line 330: Explain probe capacities.
4. Needs citations
o Lines 96-98
o Line 160: SHOM database
o Lines 187-189: This is the method of Goetz 2020 and should be cited.
o Lines 351-353: This statement either needs a citation or some elaboration as to why you hypothesize this.
5. Data sharing compliance (Field methods and historical data sections):
o I’m not sure what the benefit of providing the subset of WAV files are. In the ‘explanation of the dataset’ pdf the authors state that they couldn’t upload all of the data as supplementary which is understandable, as acoustic data is difficult to share/disseminate due to its size. However, I would suggest considering a different platform to host the acoustic data or providing the raw analyzed data so that this project could be replicable. At this point, someone wouldn’t be able to use the WAV files to replicate this project.
o The manuscript mentions adding historical data but does not clarify whether the datasets have been made publicly available. I would suggest providing repository details and ensuring datasets are accessible in accordance with the journal’s policy.
6. Adding shipping lanes (Figure 1): It would be interesting to see where the shipping lanes are in this region and how much they overlap with sperm whale habitat.
7. Add ship transects (Figure 3): This figure does not represent your sampling effort. It’s misleading because it looks as if you’ve looked everywhere in this region for sperm whales and have only found them where you have the red clouds. But this is not the case, you only sampled along the 1,500 m isobath. You need to include your sampling effort. Another reason why the historical data, which likely doesn’t have sampling effort, also dilutes your results.
8. Results in parentheses in Results section: I’m not sure how to remedy this, but having all of the results from the different statistical tests as parenthesis disrupts the flow of your results and makes it difficult to read and interpret. Instead, I might consider including all of this in a table and using an asterisk when something is significant in the text. Most of your readers won’t care what the actual values are but more about whether it was a significant result or not.
9. Figure 5: I would suggest including the standard deviation of the animal size on top of each point.
10. Fig S1.a-f: This figure is referenced in Table 2 but was missing from the material provided by the authors.
11. AB nomenclature in Figure 8: The a, b and ab nomenclature in figure 8 does not make sense to me.
12. No colors in Table 3: You mention green and red colors, but those were not apparent in my copy of the manuscript.
13. Steep slope (Lines 340-342): It’s unclear what you mean when you say “the steep slope concerns the entire coast”. Does this mean all of your data points were considered “steep”?

Experimental design

The experimental design is well thought out, but there are areas needing clarification or additional detail.
Key Issues
1. Including SST (Lines 94-96 and 384-385): You mention that SST is thought to influence distribution but didn’t include it in your analysis. Since there is access to satellite data and ocean state models, I would suggest including that analysis in your paper or not mentioning it in the introduction and again later in the discussion. If you believe your study area is too small to show differences in SST, I would make that connection distinctly in your discussion.
2. Ethical approvals (Lines 120–130): The manuscript does not specify if ethical approvals or permits for acoustic and visual surveys were obtained. I would suggest including a clear statement about ethical considerations and permits.
3. Integration of historical data (Lines 148–152): The historical data used was collected under different conditions and lack information about survey effort (both spatially and temporally). The historical data appears to be more opportunistic for me and when integrated with the data collected for this study, dilutes the strength of your findings. I would suggest explicitly discussing how historical data limitations were accounted for in the analysis.
4. Behavioral categorization (Table 1, Lines 168–169): The descriptions of behaviors lack objective definitions and could be open to observer bias. I would suggest providing specific, reproducible criteria for categorizing behaviors.
5. Example of individuals identified in a group (Lines 177-178): It would be useful to include an example of a spectrogram where several different animals were identified by their click trains as explained in lines 177-178. This figure could be supplemental.
6. Measuring the distance between two pulses (Lines 179-180): Can you cite why you chose to only measure between two pulses? Wouldn’t taking the average measurement between three pulses would be more accurate?
7. Overdispersion (Lines 206-207): It’s unclear to the reader why you check overdispersion, please elaborate why it’s appropriate for your data.
8. Bonferroni correction (Line 224): It’s unclear to the reader why a sequential Bonferroni correction was used, please elaborate why it’s appropriate for your data.
9. Explain the distinction between the two periods (Lines 230-232): Why were these two periods decided on? Are you representing seasons? Does the region have seasons since it’s the tropics? And since period 1 has more sampling effort, how will that affect your results? The sample sizes would not be equivalent.
10. Bathymetry range included in your study (Lines 345-347): You indicate differences in behavior and bathymetry, but because your survey effort was constrained, it’s important to include the potential range of depths for the animals. This probably doesn’t have to go in the discussion but should be included in the results.

Validity of the findings

The findings are supported by the data, but some claims are weakened by methodological limitations.
Key Issues
1. Unclear which results are acoustics vs. visuals: In general, it’s unclear which of your results/figures include acoustics only, visual only, and acoustics + visual and which includes data from the historical observations as well. I would suggest adding more clarifying words to the results.
2. Inconsistent treatment of aggregation size and behavior (Lines 250–264): The relationship between aggregation size and behavior is not fully explored. While statistical tests are mentioned, interpretations seem limited. I would suggest conducting further analyses or explicitly acknowledging these limitations in the discussion.
3. Bathymetry as a habitat factor (Lines 344–357 and 367-368): While bathymetry is identified as a key factor, the slope's lack of influence is not thoroughly analyzed. I would suggest exploring potential reasons for this null finding and discuss its ecological implications in more depth.
4. Areas of high maritime traffic (Lines 358-366): You mention that areas of high maritime traffic may result in more socializing behavior, but was this tested in your paper? If you’re going to mention it, it should be relevant to your study.
5. Automated methods (Lines 413-423): Your argument about why you used the manual method would be strengthened if you had applied one of the automated methods and compared. You can try CABLE by Beslin et al. which already exists and apply it to your data - http://whitelab.biology.dal.ca/CABLE/cable.htm.

Additional comments

The manuscript provides insights into sperm whale behavior and habitat use but can improve in several areas:
• Strengths:
o Use of acoustic and visual methods.
o Identification of behavioral patterns linked to bathymetry.
• Weaknesses:
o Limited discussion on integrating historical and new data.
o Insufficient clarity about ethical compliance and reproducibility.
Suggestions:
• Expand on conservation applications of the findings in the context of the Caribbean region (Lines 443–444).
• Reconsider including the historical dataset in this study as I believe it weakens your statistical strength since the data is opportunistic and does not follow the same methods as the data used in this study.
• Figures mention ‘medians and standard deviation are given’ so I would expect to see those values but I don’t. If you mean to say that the box plots represent medians and standard deviations, I would be more explicit in the figure captions.

Reviewer 3 ·

Basic reporting

This manuscript was well structured, written, and cited. There are sections in the Methods and Results that would benefit from more detail, and a few additional topics could be briefly addressed in the Discussion. I have also suggested that some of the figures could be updated or re-thought to provide more information. I have included some general comments and questions below and line by line comments and questions in the “Additional comments” section.

1) Line 288 references Fig 2a in Supplementary Material, but I was not able to find the file.

2) Can you explain why the 10 sample wav files were provided in the “Explanation of dataset” or in a general Supplementary Material document? Are the file names supposed to match up to audio files listed in the DBB_IPI_ICI spreadsheet? I searched for a few but could not find them.

3) I would suggest reviewing all references to the detection of females and/ or adult females as your IPI analysis supports the use of a “juvenile and/or adult female” class. You could include a statement early on that explains why you know or assume adult females are present because you observed social groups and/ or calves.

4) I would suggest defining “aggregation type” (and listing types) and/ or “aggregation structure” as these terms are used in various places but I was not clear on what it referred to.

Experimental design

The research objective is well defined and fills an important knowledge gap. I have no concerns about the experimental design or analysis but the manuscript would benefit from some additional detail and explanation of why certain methods were selected and any limitations. I have included some general comments and questions below and line by line comments and questions in the “Additional comments” section.

1) I would suggest including a paragraph in the Discussion about the sampling design and decision to focus on the 1500 m isobath on one side of Martinique. What was the justification and how might this impact your results? Did the historical data sample a larger area and find that sperm whales were most commonly detected around this contour? Is there significant vessel activity in a specific area?

2) How does your sampling method support how you defined aggregations and your analysis of behaviours? A brief explanation in the methods about how sampling duration (visual and acoustic) at each acoustic point was determined would help. Were the individuals used in your analysis representative of the individuals in the area (i.e., is it possible individuals were missed visually or acoustically and how would this impact your results?). How were the visual observations linked to the acoustics (i.e. is it possible you included acoustic signals from an individual you did not detect visually given the acoustic detection range of sperm whales is >1 km)? Is visibility from a 6.5 m boat (i.e. close to the waterline) often >1 km? How did you determine whales were within 1 km? A few additional details in the Methods would answer most of these questions.

3) The combination of the historical data and your data for the analysis of habitat use should be explained in more detail, and any caveats should be briefly addressed in the discussion.
A brief explanation for why those statistical tests and parameters were selected would be helpful. For example, why was the presence of an immature animal tested but not the presence of an adult male?

4) Why was bathymetry and slope tested but not latitude or position North to South?

5) A brief explanation of how the two periods were defined and why would be helpful, given you interpret this as being related to male migration.

Validity of the findings

This research is valid, and the impact and potential for future research is discussed.
Given this is new and important research, I encourage the authors to emphasize some of the simple findings in regards to sperm whale presence and demographic composition. For example, the detection of adult males with social groups or what area offshore of Martinique might represent important habitat (e.g., where relative density is highest). Updating a few of the figures would also help readers interpret some of the basic results easily.

As risk of vessel activity is mentioned several times, I would also encourage a more in depth discussion of how sperm whale activity in the study area may be impacted by vessels and/ or how these results specifically could help mitigate risk.

Data is provided in the supplementary materials but the inclusion of 10 wav files could be explained.

Additional comments

Below are my line by line questions and comments:

Line 26-27: “...aggregations consisted mainly of females and/or juveniles, immatures, often joined by mature males, …” This wording is a bit confusing and would benefit from rephasing. Were aggregations (as defined / reported in this manuscript) joined by adult males or did they include adult males? Furthermore, did you group females and juveniles, or adult females and immature individuals?

Line 30-31: Further clarification or detail in the discussion would help justify the conclusions presented in this sentence. Perhaps this should say “bathymetry and/or proximity to coast” as they were not tested independently? What are the temporal dynamics of aggregation structures? Does this refer to the decrease in mean AS between the two periods defined in your analysis? Perhaps the manuscript should define “aggregation structure” and/or “aggregation type” so the results and discussion related to this sentence are easy to find in the text.

Line 34-35: A paragraph in the discussion about how these results could inform vessel traffic management and/ or reduce the risk of vessel strike would be helpful as right now it feels like an afterthought.

Line 42: Can you provide any other references that describe vessel strike risk to sperm whales? Laist et al., 2001 seems to provide historical data (pre-2000) for multiple species and from a large region. Is there any evidence that sperm whales are at risk in the waters off of Martinique? I ask because you have highlighted the risk of vessel strike and potential mitigation tools in your abstract and introduction.

Line 43: Is this meant to say “...probable limited probing abilities…”? Could you briefly explain what “probing abilities” refers to?

Line 59: Can you add a reference for the function of the spermaceti organ impact on dive depth and duration?

Line 68: Suggest changing “...sperm whales spend majority of their time..” to “...sperm whales spend the majority of their time..”

Line 71: Is Zimmer 2003 or Jaquet 2001 the primary reference for the ICI of regular clicks?

Line 72: Creaks are produced while searching for prey or during prey capture attempts, not just “during the capture of prey”. Suggest rephrasing.

Line 75-77: Codas “... represent a clan signature that groups several social units within the same large geographical area, thus distinguishing sperm whales from different oceans.” I am not sure this explanation of clan signatures can be applied to all codas for all sperm whales, perhaps this sentence could be rephrased for clarity?

Line 78-87: Those who are not familiar with sperm whale clicks might benefit from a brief explanation of the multi pulse structure of sperm whale clicks before the introduction of IPI. Some of the sentences in this paragraph would be rephrased to be more clear for those who are unfamiliar with sperm whale physiology or the bent horn theory. For example, “...a small portion of the sound is emitted directly in the center (pulse p0)...” - does this refer to the p0 being emitted from the center of a sperm whale’s nose?

Line 86: I would suggest removing the word “accurate” or defining what the means, as an animal’s recorded IPI can vary and the equations used to convert IPI to body length have limitations and caveats.

Line 95: I believe this sentence is missing a word: “...with groups found in colder waters than solitary individuals…”

Line 101: suggest deleting the "...whereas acoustics…” or rephrasing. Seems to be a word missing after “...sperm whale clicks…”

Line 107: suggest rephasing “... the importance of culture on environmental parameters…” as sperm whale culture would not change or impact the environment, instead environmental parameters might be a factor in culture?

Line 113: Does ICI calculation refer to the analysis of codas?

Line Line 130-131: Would suggest reversing the order of these sentences, to explain the sampling design (10 data collection points), and then explaining how it was decided not to sample. A brief explanation for how this sampling design was decided on would be helpful.

Line 132: If sperm whales were detected, did you move off the acoustic point to try to find them visually? If so, how far from the acoustic point would you travel? If sperm whales were not detected acoustically within 5 minutes, you moved to the next acoustic point?

Line 133: I assume you mean the minimum distance between your vessel and a sperm whale was 100 m?

135: Was 5 - 10 minutes of acoustic recordings a predetermined duration or did you collect data until you had recorded all the individuals you detected visually? This is a fairly short period given sperm whale foraging dive duration. A brief explanation of how the sampling method was decided on and any limitations in the discussion would be helpful.

Line 137-142: Was all the acoustic data collected at a sample rate of 96 kHz?

Line 148: Is there a citation for the historical data? How was this data used in your analysis of the distribution of behaviours? Can you provide a table summarizing the data in the supplementary materials?

Figure 1:
It would be standard to label the island on the map
Suggest labelling the places referenced in line 125-126 (Cap Salomon, Le Precheur)
Suggest mapping locations of historical data collection as they used used to decide on sampling the 1500 m isobath

Line 158-159: Is there a reference for how slope was categorized?

Line 160: Is there a reference for the SHOM database?

Line 163: Is where a reference for definiting an aggregation as all whales observed within 1 km? I assume this means they were observed visually as you did not conduct acoustic localization. How were the visual observations linked to the acoustics (i.e. is it possible you included acoustic signals from an individual you did not detect visually given the acoustic detection range of sperm whales is >1 km)? Is visibility from a 6.5 m boat (i.e. close to the waterline) often >1 km? How did you determine whales were within 1 km?

Line 167: Was the GPS point taken for an aggregation different from the acoustic point?

Line 168: How long did you observe aggregations in order to define behaviour? How many visual observers were on the vessel to track the 1 km radius around the vessel?

Table 1: The information in the “Visual” column seems incomplete for every behavior. Please add some more detail. For example, did hunting involve anything other than resting on the surface? I assume it included observation of a fluke up and / or dive of a certain duration? For socializing, what is “jump…”? For travelling, what is a medium or fast velocity?

Line 174: Why was 10 kHz selected as the frequency maximum? Why does Figure 2 include a spectrogram with a 48 kHz maximum?

Figure 2: Those not familiar with sperm whale clicks and IPI may benefit from some additional labelling. For example, you could label each click something like “Whale 1 click” and “Whale 2 click”, and/ or the p1 and p2 pulses of each click.

Line 183: Does “prolonged first impulse” refer to the p0 or p1?

Line 184: Calves were noted visually but their presence in aggregations was not included in the results?

Line 188-190: How did you determine which equation to use to calculate Animal Size? Was it based on an average IPI value of <=4 ms or >4 ms? Did you consider testing the equation presented by Ferrari et al., (2024) (https://www.nature.com/articles/s41598-024-51194-5) for IPI <2 ms?

Line 202: How was it decided the mean size would be a good parameter?

Line 203-204: This sentence does not make sense and could be rephrased: “... for each existing aggregation size for which (2 to 5), the previous means and standard deviation were averaged.”

Lines 205-240: sample sizes of different groups used in statistical tests are reported in the methods, before these results are summarized. For example, the number of aggregations with an immature animal versus without is reported here before the number of aggregations detected is reported. If it makes sense to leave these sample sizes here, I would suggest including a table that makes it easy to relate these sample size values to the total number of aggregations or individuals and to the results section. For example, N adult male = 28, N adult female/juvenile = 30 and N immature = 16 equal 74 detections. N hunting = 27, N moving = 76, N resting = 12, and N socializing = 10 equals 125 detections. If these include 71 data points from historical data, that leaves 54 detections from your data? Why are the sample sizes different when analyzing slope? A table with a column for notes may make this easy to explain and interpret.

Line 213: suggest changing “recalled” to another word such as “tested” or “evaluated”

Line 216: suggest rephrasing or replacing “according to” with something like “in relation to”. I am not an expert in statistics, is it common to conduct these tests using environmental factors as the dependent variable and your observations as the independent variable?

Line 230: Is aggregation type ever defined? Were aggregations put into categories? Is it based on the number of individuals or the composition?

Line 230-232: How were these two periods selected? Was it based on environmental factors, effort, number of detections, etc.?

Line 233-234: What do these sample sizes represent? Number of surveys conducted during each period? Does Nperiod1 = 18 or 17? How did the analysis of slope relate to the survey period?

Line 242: I suggest making it clear that the number of surveys equals days surveyed and reported number of days data were collected during each month and each period. I would also suggest summarizing days with sperm whale detections here, before presenting relative density. Is it not relevant at which acoustic points sperm whales were detected? Results seem to be reported at the survey / day level.

Line 247: How was relative density calculated? Do you have a reference? This is not explained in the methods. How did you use the historical data? Was it grouped into aggregations to match your data? How were the categories of low, moderate and high density categorized? Perhaps rephrase or add more explanation to justify the statement that the map shows a “pronounced distribution between the 1000 m and 1500 m isobaths” as you sampled at the 1500 m isobath and the map seems to show density between 1000 - 2000 m. Given relative density is not discussed further a simple map that shows points instead of relative density might be more helpful. Points could also show behaviour, to supplement the analysis of behaviour at different bathymetries.

Figure 3: I suggest including a map with points for detections (or aggregations), distinguished based on whether they are your data or historical data.

Line 249: How many individuals could not be acoustically characterized?

Figure 4: What does “release date” refer to in the title? Another way to present this data could be to have a bar plot with the number of individuals per aggregation, grouped by date. For example, 12/01 would have 4 bars representing the 4 aggregations, with the y axis / height of the bar indicating the number of animals in the aggregation.

Line 252: Aggregations can include 1 whale? The methods state IPI was cut off at 2 ms, but here a IPI of 1.9 ms is reported.

Line 255: I suggest stating that you are assuming adult females were detected because you observed social groups with calves. IPI and AS can be used to identify a body length that might be an adult female or a juvenile animal (male or female), so a certain AS does not confirm that an adult female was detected. I suggest reporting the number of detections in each class here, as it is only in the figure title below.

Line 257: It would be helpful to see a visualization of the AS grouped by aggregation to see the distribution of the data (both number of animals per aggregation and how the AS is distributed). Figure 6 could be updated to show this (for example, a scatterplot with aggregation number on the x axis and AS on the y axis) and include the AS mean and SD for each aggregation, thereby illustrating the same point as it does now but with more information shown.

Line 260: Does this statement refer specifically to the larger aggregations: “This pattern highlights the presence of individuals of all age classes…”? From my understanding, what you have illustrated in Figure 6 is that larger aggregations have greater variability in AS, indicating they may have individuals from different classes.

Figure 6: Line 252 states aggregation size ranges up to 9 animals, but this figure only goes to 5.

274: A simple presentation of behaviour results would be helpful, such as a table with behaviour, number of aggregations and/ or number of individuals, and which dataset (historical vs. your data).

Line 277: I suggest changing “higher bathymetries” to “greater” or “deeper”

Table 3: the title refers to colours but there are none in the table.

Line 283: Doesn’t period 2 include the data collected in April?

298: Does the sentence “the identified sperm whales were more likely to be …” refer to all the sperm whales detected in aggregations other than the one lone male detected? Perhaps remove the “more likely to be” from the sentence.

Line 300: Suggest including “juvenile and/or adult female”

Line 306-309: Your results do not report how often an adult male was detected in an aggregation, but you refer to results from previous studies. Would this be an easy thing to report?

Line 311: It is not clear how this sentence relates to the rest of the paragraph or your analysis and the wording should be clarified as you have not calculated proportions of juvenile and adult females, or proportions of immature males. You have categorized animals in three classes (immature, juvenile/ adult female, or adult male).

Line 312-313: What is the percentage of adult males in your study area? It would be helpful to report the values found by the studies referenced and why the results are comparable (what is the data based on and how was percentage determined?).

Line 314: The sentence here begins with “Similarly…” but seems to have no relation to the information presented before it. Jaquet (2006) found a difference in average body length between regions, but that is not relevant to the proportion of individuals from different demographic classes.

Line 317: This is not a typical use of the term “minority”, I suggest finding a different word or rephrasing

328: I am still not clear on what “types of aggregations” refers to, as the analysis tests characteristics one at a time. Perhaps you could explain the types of aggregations in the methods or results.

328-332: This explains why the statistics tested the presence or absence of an immature animal. A sentence about this in the introduction and/ or methods would help the reader understand why that was tested. Can you also explain why testing for presence or absence of an adult male was not included?

331: Suggest replacing “small” with “calves” or “young”

340: Suggest replacing “concerns” with another word such as “characterizes” or “represents”

344: A brief explanation of why bathymery and slope were selected to analyze distribution, and why other factors such as distribution north to south (given the relative density map shows peak density around the 3rd most northern acoustic point) were not assessed. This seems especially relevant given the results may inform management of vessel activity.

Line 344-347: A brief discussion of any limitations or caveats would be helpful. It is not clear how long behaviour was observed or how the historical data was integrated with your results. The median bathymetry for the different behaviours varies by <500 m, are you confident the bathymetry value associated with each behaviour was accurate? A map with aggregation GPS points and a 1 km buffer would make it easier to assess how accurate a single bathymetry value would be for each aggregation and the observed behaviour.

Line 353: The last part of this sentence does not seem related to the rest of the paragraph, more explanation would be helpful

Line 358-359: The segway to maritime traffic could use some more explanation. Are you suggesting that there is strong social cohesion throughout your study area because there is maritime traffic throughout?

Line 369-372: How does topography differ from your results regarding bathymetry and slope? What is topographic distribution? Is this a combination of slope and bathymetry? This is a great place to discuss how the historical data was combined with yours and any other differences in the methods.

Line 376-382: A brief discussion of how the two periods were defined and why would be helpful, as well as any caveats such as how results might differ if data had been combined differently. Would it be more simple to test the presence or absence of adult males between these periods instead of using average size of individuals?

Line 378: Important to be clear that you are assuming aggregations included adult females and/or juveniles, or state why you know females were present (e.g., visual observations).

Line 388: Suggest changing “Since” to “As” and “hunting” to “underwater” or “searching for prey”

Line 405: Your sample size for socializing behaviour is 10, and at line 374 that is the sample size from the historical data. Were these two codas used to identify socializing in your results? Please include more detail (e.g., a table) in your results to show how many times each behaviour was recorded, divided by dataset (historical vs. your data).

Line 423: A sentence about any limitations of calculating Animal Size based on average IPI using the two equations and defining class based on this estimated size would be beneficial.

Line 438: Suggest changing six months to five months

Line 439: Suggest changing “in Martinique” to “off Martinique” or “in Martinique waters”

Line 440: This is a pretty broad statement, I suggest focusing on the effectiveness of your methods for studying sperm whales or explaining how it applies to all cetaceans

442: The reference to the Repcet(R) system feels like an afterthought. A few more sentences about how and why your method and results can be used to mitigate vessel strike risk off Martinique would be beneficial.

---

## Round 0.2 · Minor Revisions

Both reviewers agree that the paper has been significantly improved, but some issues still need to be addressed. Specifically, please pay close attention to:

Reviewer 1’s comments on clarifying the model specifications in the Methods section.

Reviewer 2’s suggestions for improving clarity in both the Methods and Conclusions sections.

Additionally, please refine the language throughout the paper, as some of the concerns stem from unclear phrasing.

Reviewer 1 ·

Basic reporting

The English used in the manuscript is generally clear and unambiguous, though there are some minor grammatical and formatting inconsistencies, such as the alternating use of “Figure” and “Fig.” which should be standardized. The overall structure of the manuscript has been significantly improved, particularly with the addition of the “Implications for Conservation and Management” section, which strengthens the relevance of the study for both scientific and conservation audiences. It is also recommended to incorporate geographic coordinates into the maps to facilitate spatial orientation and interpretation.

Experimental design

As I mentioned in the first round of revision, more details are needed regarding the models used. For example, it should be specified which GLM package was used, how interactions between variables were tested or included, and how model selection or validation was performed. Including this information would improve the reproducibility and clarity of the statistical analyses.

Validity of the findings

no comment

Additional comments

no comment

Annotated reviews are not available for download in order to protect the identity of reviewers who chose to remain anonymous.

Reviewer 2 ·

Basic reporting

No comment

Experimental design

No comment

Validity of the findings

No comment

Additional comments

No comment

Reviewer 3 ·

Basic reporting

The authors have made a significant effort to address the first round of comments and I appreciate their work. I do have additional questions / comments which I hope will be easy to address.

The writing could be more concise and clear. There are many run-on sentences and irregular use of commas or words. I have identified some in the line by line comments. Several paragraphs in the Discussion would be more impactful if rearranged and simplified.

The various sample sizes are still difficult to track and clear reporting in the methods and results would be helpful. Based on the sample sizes reported for your study, there were 72 (or 74?) acoustic samples used for IPI analysis, 21 observations of acoustic behaviour, and 24 visual observations? The supplementary tables in the “Answers_to_comments” document are helpful but the sample sizes should be easier to interpret. I suggest including one table with the total sample size from each data type (your study IPI analysis, acoustic behaviour, visual behaviour, historical data, and whale watch data) and/ or adding the sample size totals to the tables so it is easy to link the data to the sample sizes reported in the figures. It would be helpful to clearly report the observations that were included in your analysis, for example 69 observations from historical data and two from watch watching data (N = 71).

The number of individual sightings (or acoustic detections?) is reported as 72 (abstract, Line 319), however in multiple places (Fig 4, 5 and 9, Line 309, Line 329) the sample sizes reported add up to 74. I am still not clear on how acoustic and visual observations were combined.

Table S4 reports a total of 45 observations of behaviour from your study, including both visual and acoustic data. Table S1 “Summary of sample sizes in habitat use analysis” (provided in the Answers_to_comments document) reports 54/55 observations from your study. Figure 3 reports 26 observations “during the study”, this does not add up to the number of aggregations or any of the individual counts.

The abstract reports 19 aggregations. Line 302 and 304 reports the number of aggregations in period 1 as 17/18 and period 2 as 7, putting the total at 24/25. Should this be the number of surveys / number of days?

The legend on Figure 1 reported “all sperm whales observations (N = 188)” while the legend on Figure 3 reports 26 observations “during the study” and 162 additional observations (total = 188). Line 169 reports 143 observations from the Aquasearch historical database, 69 with GPS position and behaviour. Line 175 reports 19 observations from whale watch data, 2 with behavioural observations. This is consistent as 143 + 19 = 162 “additional observations” however it is not easy to interpret.

Table S1 “Summary of sample sizes in habitat use analysis” (provided in the Answers_to_comments document) reports 68/69 observations from historical data, 2 observations from whale watching data, and 54/55 observations from your study. Table S3 reports 54/55 observations of behaviour from your study and 70/71 observations from “additional data” (historical + whale watcher). Line 287 reports behaviour was analyzed according to bathymetry using 126 observations but behaviour was analyzed according to slope using 124 observations. Adding up the right data in Table S3 provides these totals, but it takes extra effort by the reader and the number of observations you collected (IPI, acoustic behaviour, visual behaviour) should be consistent and clear to the reader.
I could not find the document/file with the supplemental materials (Table S1-4) included in the “Answers_to_comments” document. Please note in the document different tables have the same table number (multiple Table S1).

Perhaps it is not significant, but the results do not provide basic information like the proportion of aggregations that included a certain number of animals (e.g., X% including less than 4 animals) or the breakdown of size classes (e.g., just immatures, just adult males, adult males and juvenile/adult females, etc.). One percentage is reported in the abstract (“37% of the aggregations were made up of females and/or juveniles, immatures, with a mature male nearby”) and another in the Discussion (“presence of mature males in nearly 80% of the characterized aggregations”). I would suggest reporting these percentages in the results or in a table. It is a bit unusual to report a value in the abstract that is not referenced anywhere else in the manuscript, I would expect this is a highlight of your results and discussion.

Experimental design

I am still not clear on how aggregations within 1 km were established. Animals were not localized acoustically and a 8 km range is mentioned. See line by line comments under “Additional comments” for areas where clarification would be helpful. As you analyze the number and size of animals in aggregations, it is important that the reader can understand how data were combined spatially. Furthermore, if the same location (GPS point) was used for a behaviour observed visually and one observed acoustically - and used to conduct a statistical test of behaviour according to bathymetry or slope - it is important for the reader to understand how accurate that location may be. If there is uncertainty, that can be acknowledged in the discussion.

Validity of the findings

The incorporation of visual and acoustic data from your study and different datasets adds complexity and potential limitations that are not fully addressed in the methods or discussion. The authors have added important discussion at Line 535 but I am not an expert in statistics and do not know how variations in effort and methods may have impacted the statistical tests and interpretations. Variations in effort (spatial and seasonal if historic data were collected at different times of year) and whether this could impact conclusions regarding sperm whale distribution and behaviour are not discussed. The analysis of behaviour according to bathymetry and slope may be impacted by the different sample sizes from historic versus study data and from acoustic (hunting and socialising) versus visual (moving and resting) observations. This could be acknowledged, if not briefly explored, in the discussion.

The conclusions regarding average size of individuals by aggregation size would benefit from some clarification. You state that “as aggregation size increased, the difference in size among individuals within these groups also increased” (Line 334) and “As the size of the aggregation in our study increased, the AS standard deviation between individuals increased, indicating the presence of individuals with more size differences in larger aggregations” (Line 372), however Figure 6 displays an increase in standard deviation with aggregation size up to 5 individuals, and then a decrease between aggregation sizes of 6 - 9 individuals. Your interpretation and conclusions do not seem consistent with the data. In addition, your estimated aggregation size seems to be based on the number of click trains for which an analysis of IPI could be completed and it is not clear from the methods how confident one should be in the total aggregation sizes estimated. If your conclusions are based on sample size (number of aggregations with that number of animals), perhaps you could make this clear and/or include sample size on Figure 6. Table S1 (Social structure of each aggregation acoustically characterised) reports only a few aggregations with more than 5 individuals.

Additional comments

Table 1: The column heading “Number of adult females/juvenile males” should be updated to “Number of adult females/juveniles” as the juveniles could be male or female.

Line 44: Remove period after “maritime traffic”

Line 50: Without evidence of vessel strike or mortality in your study area it may be premature to suggest maritime traffic may “partly explain the decline in sperm whale numbers”. If this is not your intention, you could include which populations/regions Gero & Whitehead and Rinaldi et al., studied so it is clear to the reader you are reporting their results / interpretations.

Line 84: Suggest changing to “directly from the center of the nose (p0)” as this sentence would not be clear to those unfamiliar with sperm whale physiology and sound production.

Line 99: Missing the “more”

Line 144: Would suggest rephrasing to something like “sea conditions worsened with distance from the coast”.

Line 149: How did you determine that sperm whales were up to 8 km away? Or do you mean in theory your acoustic equipment would detect sperm whales 8 km away? Either way, your definition of aggregation (Line 191) is not consistent with this range.

Line 193: To determine the range of sperm whales detected visually and acoustically you would need to localize each individual or have a somewhat precise coordinate for each individual. I do not understand how QGIS was used to determine that all animals were within 1 km. In addition you refer to sperm whales detected acoustically being 8 km away (Line 149).

Line 205: “Characterised” is not the correct term here, suggest using another word such as “detected” or “recorded”

Line 207: I have never seen the term “apneic” used to describe the cessation of breathing by marine mammals while underwater. In this sentence it may be more relevant to explain that sperm whales remain below the surface for long periods as that is what impacts visual detection.

Line 233: Would it be easy to include the average IPI range each equation was used for? As at higher average IPI values, either equation would result in an estimated body length above 11 m. For example, other studies have reported applying Gordon (1991) to IPI values less than 4 ms and Growcott et al (2011) to IPI values greater than 4 ms.

Line 238: Suggest changing to “juvenile and/or adult female”

Line 286: Were historical observations without GPS data “not removed from analysis”? Given you had 143 observations from Aquasearch but reported including 68-69 in the analysis, it seems that more data were excluded.

Line 251: Do you have a reference for cooperative hunting among sperm whales?

Line 302: The abstract reports 19 aggregations. Line 302 and 304 reports the number of aggregations in period 1 as 17/18 and period 2 as 7, putting the total at 24/25. Should this be the number of surveys / number of days?

Line 312: Remove “numbers of” from “24 numbers of days of data collection”

Line 315: Suggest revising “pronounced distribution” to something like “concentrated distribution”

Line 330: Suggest rephrasing as “certain to ensure parental care” is an unusual phrase, for example “The presence of adult females and immature individuals suggests parental care”.

Line 333: Figure 6 reports average size of individuals ranging from 9.3 - 11.5 m (not 10.6 - 11.5 m).

Line 337-338: These sentences could be combined into one sentence.

Line 347: Missing “more” after the bracket

Line 349-351: No difference between behaviours or no significant difference?

Line 360: Do you have a reference?

Line 361: The structure of the aggregations were consistent with what? Do you mean all the aggregations you analyzed consisted of a similar number or composition of individuals?

Line 362: The last part of this sentence (use of “with the proximity of”) could be clarified.

Line 371: This sentence is still not quite clear, it makes it seem like all animals in aggregations of two or more were immatures, juveniles or females (no adult males).

Line 372: This conclusion is not consistent with the data shown in Figure 6.

Line 386: Suggest revising to “juveniles male/females, adult females, ...”

Line 387: The percentage of adult males in your study area is still not reported anywhere, if it is part of the discussion and compared to other studies I would expect a value to be reported.

Line 429: Do you have another reference for this? Do giant squid inhabit the waters off Martinique?

Line 438-467: Discussion of behaviour and aggregation type in relation to spatial distribution is difficult to follow in these paragraphs. For example, submarine canyons and squid (i.e., hunting) are referenced in three paragraphs when this topic could be covered in one sentence. You suggest sperm whales are targeting canyons but canyons have not been mentioned in the description of the study area.

Line 456-459: This sentence could be simplified, it is difficult to understand your point.

Line 463: I would expand on this to discuss how this might impact your results regarding behaviour according to slope or bathymetry. Or acknowledge it may be a factor in those results and conclusions.

Line 465: This sentence does not make sense and does not seem related to the rest of the paragraph. Is this referencing the fact that you may have missed individuals in your study area/ an aggregation because they were diving?

Line 466: Why specifically the 2000 m isobath? Your range is restricted in all directions especially if you are relying on acoustics and animals are not calling.

Line 471: Suggest rephrasing to “aggregations likely/possibly/may have consisted of more females/young males…”

Line 483: Suggest removing “necessarily”

Line 486: “Bathymetries deeper than 2000 m and 2500 m” or between 2000 - 2500 m?

Line 505: What is a “neutral methodology”? I am not clear on how that heading applies.

Line 537: Suggested adding "N total = 10"

Line 544: Suggest changing “stayed in front of each group” to something like “observed each group”

Line 552: Suggest adding “data” after acoustic and revising the use of “level of precision” as it does not seem to apply

Line 554: Suggest rephasing the first section to something like “is not consistent with sperm whale habitat”. Do you have a reference for maritime traffic or can you explain how you know this?

---

## Round 0.3 · accepted · Accept

The authors have tackled all the suggestions made by the reviewers and me. The paper is ready to be accepted. Nevertheless, some minor edits of language should be undertaken

For example, replace "No significant correlation" with "No correlation" in Table 4 and similar usage elsewhere in the manuscript.